# Pancreatic Exocrine Insufficiency and the Gut Microbiome in Pancreatic Cancer: A Target for Future Diagnostic Tests and Therapies?

**DOI:** 10.3390/cancers15215140

**Published:** 2023-10-25

**Authors:** James M. Halle-Smith, Lewis A. Hall, Sarah F. Powell-Brett, Nabeel Merali, Adam E. Frampton, Andrew D. Beggs, Paul Moss, Keith J. Roberts

**Affiliations:** 1Hepatobiliary and Pancreatic Surgery Unit, Queen Elizabeth Hospital Birmingham, University Hospitals Birmingham NHS Foundation Trust, Birmingham B15 2GW, UKj.k.roberts@bham.ac.uk (K.J.R.); 2Cancer and Genomic Sciences, University of Birmingham, Birmingham B15 2GW, UK; a.beggs@bham.ac.uk; 3Institute of Immunology and Immunotherapy, University of Birmingham, Birmingham B15 2TT, UK; 4Section of Oncology, Department of Clinical & Experimental Medicine, University of Surrey, Guildford GU2 7WG, UKadam.frampton@surrey.ac.uk (A.E.F.); p.moss@bham.ac.uk (P.M.); 5Minimal Access Therapy Training Unit (MATTU), Leggett Building, University of Surrey, Guildford GU2 7WG, UK; 6Department of Hepato-Pancreato-Biliary (HPB) Surgery, Royal Surrey County Hospital, Egerton Road, Guildford GU2 7XX, UK; 7Colorectal Surgery Department, Queen Elizabeth Hospital Birmingham, University Hospitals Birmingham NHS Foundation Trust, Birmingham B15 2GW, UK

**Keywords:** pancreatic cancer, microbiome, pancreatic exocrine insufficiency, pancreatic enzyme replacement therapy

## Abstract

**Simple Summary:**

It is now thought that bacteria in the gut and tumour of patients with pancreatic cancer play an important role in the growth of the cancer and its sensitivity to chemotherapy. It has been shown that pancreatic exocrine insufficiency (PEI), common in pancreatic cancer patients, is linked to poorer outcomes. Conversely, it has also been demonstrated that treatment with pancreatic enzyme replacement therapy (PERT) can improve survival in pancreatic cancer patients. The reasons for this are not fully understood, but it is possible that PEI alters the gut and tumour microbiome of pancreatic cancer patients towards a less favourable composition, whereas PERT can reverse these changes and make the gut and tumour microbiome more favourable. If true, this could represent an opportunity for the development of new diagnostic tests and therapies for pancreatic cancer, which remains one of the deadliest cancers.

**Abstract:**

Pancreatic exocrine insufficiency (PEI) is common amongst pancreatic cancer patients and is associated with poorer treatment outcomes. Pancreatic enzyme replacement therapy (PERT) is known to improve outcomes in pancreatic cancer, but the mechanisms are not fully understood. The aim of this narrative literature review is to summarise the current evidence linking PEI with microbiome dysbiosis, assess how microbiome composition may be impacted by PERT treatment, and look towards possible future diagnostic and therapeutic targets in this area. Early evidence in the literature reveals that there are complex mechanisms by which pancreatic secretions modulate the gut microbiome, so when these are disturbed, as in PEI, gut microbiome dysbiosis occurs. PERT has been shown to return the gut microbiome towards normal, so called rebiosis, in animal studies. Gut microbiome dysbiosis has multiple downstream effects in pancreatic cancer such as modulation of the immune response and the response to chemotherapeutic agents. It therefore represents a possible future target for future therapies. In conclusion, it is likely that the gut microbiome of pancreatic cancer patients with PEI exhibits dysbiosis and that this may potentially be reversible with PERT. However, further human studies are required to determine if this is indeed the case.

## 1. Introduction

### 1.1. The Human Microbiota

The human microbiota is a term used to describe the approximately 100 trillion microorganisms present in each individual [1,2,3]. With the emergence of more sophisticated bacterial genome sequencing [4,5,6,7], there is increasing interest in the interactions between the human microbiota and the human host. Such sequencing techniques include organism-level profiling with 16s rRNA gene sequencing or the more sophisticated gene-level profiling possible with metagenomic and metatranscriptomic sequencing [4,5,6,7]. These sequencing techniques have allowed a greater understanding not only of the composition of these microbial communities, but also an insight into their function [4,5,6,7], and the role they may play in the wider metabolic and immune function of the host [8].

### 1.2. The Gut Microbiome

A microbiome is defined as a population of microorganisms in conjunction with their genes in a specific body system; as such the microbial community in the bowel is referred to as the gut microbiome [1,2,3]. It has been demonstrated that the gut microbiome plays a key role in modulating host metabolic and immune function [9]. Within the gut, it has been demonstrated that the gut microbiome plays an important role in the development and progression of inflammatory bowel disease [10,11,12] and colorectal cancer [13,14]. Recent evidence highlights a functional role of the gut microbiome not only in cancer development and progression but also in defining the efficacy and toxicity of chemotherapeutic agents [15]. Beyond the gut however, it is thought that the gut microbiome is also linked to the progression of certain conditions such as chronic liver disease [16,17] and pancreatic ductal adenocarcinoma (PDAC) [18,19]. In terms of mechanisms, it is thought that bacteria and their metabolites exert their effects through translocation via the portal circulation, mesenteric lymph nodes or directly through the biliary and pancreatic duct systems [20,21,22]. All this combined means that the gut microbiome is increasingly viewed as a potential route for future diagnostic tests and a therapeutic target for many different conditions, including PDAC.

### 1.3. Pancreatic Ductal Adenocarcinoma

Whilst progress has been made in the treatment of other major malignancy types, PDAC remains one of the leading causes of cancer death worldwide, with 5-year survival rates persistently below 10% [23,24,25]. One reason for this poor survival is the tendency for PDAC to present late, meaning that many patients already have involvement of nearby major vascular structures which can preclude curative resection [26]. Similarly, many PDAC patients have distant metastatic disease at presentation, with the majority being hepatic due to the portal drainage of the pancreas [27,28,29], and current international practice guidelines advise against resection of PDAC in the setting of liver metastases [30,31,32,33,34]. Should the tumour be resectable and the patient receive systemic therapy, 5-year survival can reach up to 30% [35].

Even when surgical resection is achieved, there is a high likelihood that the patient will experience disease recurrence [36]; 75% of patients who undergo surgical resection and adjuvant therapy for primary PDAC will experience recurrence within 2 years [23,37,38]. As a result, there is an urgent need for new diagnostic and therapeutic targets in this challenging disease. The gut microbiome may represent one of the more promising prospects for future therapies.

### 1.4. Changes in the Gut Microbiome in PDAC Patients

It is increasingly appreciated that the gut microbiome is linked to the evolution of PDAC, with a recent study identifying changes in the gut microbiome that are present early in the disease process [19]. Further evidence that the gut microbiome may play an active role in the progression of PDAC comes from differences reported in the composition of the gut microbiome between short-term survival (STS) and long-term survival (LTS) PDAC patients, with greater diversity being associated with a more favourable prognosis [39].

### 1.5. Pancreatic Exocrine Insufficiency in Pancreatic Cancer

Pancreatic exocrine insufficiency (PEI) is defined as insufficient secretion of pancreatic enzymes and bicarbonate to maintain a normal digestion [40]. PEI has been shown to be both prevalent and progressive amongst patients with pancreatic cancer. A recent prospective cohort study used the ^13^C mixed triglyceride breath test to investigate the prevalence of PEI amongst 254 patients awaiting pancreatoduodenectomy for an oncologic indication [41]. This study reported an incidence of 32.8% in those awaiting resection [41]. In addition, a systematic review in 2016 identified four studies evaluating PEI preoperatively, all using faecal elastase (FE-1), and found a median prevalence of 44% (range 42–47%) [42]. This results from lower enzyme secretion due to tumour burden as well as reduced enzyme delivery and activation as a consequence of pancreatic duct obstruction, which prevents the passage of enzymes and their accompanying bicarbonate-rich fluid into the duodenum [43]. Following pancreatoduodenectomy, there is also significant disruption to the physiological mechanisms by which pancreatic secretion is controlled, with some data indicating that the method of reconstruction may influence the degree of postoperative endocrine and exocrine pancreatic function [44,45].

Untreated PEI has a significant impact on quality of life and leads to a range of mentally and physically distressing symptoms including frequency, urgency, bloating, diarrhoea, fatty stool flatulence, loss of appetite and vomiting [46,47]. A major consequence is maldigestion resulting in malabsorption, malnutrition, and the ensuing nutritional deficiencies (including albumin, pre-albumin, transferrin, lipoproteins, fat soluble vitamins, calcium, magnesium, zinc, thiamine, and folic acid [48,49]). Clinically, PEI is associated with an increased risk of osteoporosis, sarcopenia, and cardiovascular events [50,51,52]. In pancreatic cancer, sarcopenia has been associated with increased perioperative mortality and reduced overall survival. There are multiple neural and hormonal factors that influence sarcopenia in PC and many of these also have interactions with the gut microbiome [53,54]. PEI has also been shown to increase length of stay, post-operative complications and costs following pancreatic resection [55,56]. It is also likely that PEI associated with PDAC leads to important changes in the gut microbiome which, if more clearly understood, could lead to the identification of new diagnostic and therapeutic targets.

### 1.6. Pancreatic Enzyme Replacement Therapy

Pancreatic enzyme replacement therapy (PERT) refers to the oral administration of capsules containing commercially produced digestive enzymes, namely lipase, protease and lipase. Treatment of PEI with PERT has been shown to improve quality of life [57,58] and also significantly increases survival in resectable and unresectable pancreatic cancer patients [59,60]. This has led the United Kingdom’s National Institute of Health and Care Excellence (NICE) to recommend PERT for pancreatic cancer patients [61,62] without the need for a positive diagnostic test. This was supported by UK specialty consensus guidelines release in 2021 [62]. The underlying basis for the beneficial effects of PERT are not fully understood, but contributing factors appear to be related to the ability to alleviate malabsorption [63,64], which allows patients to tolerate physiologically demanding treatments such as major pancreatic resection and/or chemotherapy. On a micronutrient level, the absorption of fat-soluble vitamins appears crucial. For example, Vitamin D modulates the function of stromal cells (e.g., fibroblasts) in patients with PDAC [65] and deficiency is associated with poorer survival in advanced pancreatic cancer [66]. Recently, interest has been increasing in the intestinal microbiome as a modulator of many diseases (e.g., inflammatory bowel disease) although how PEI may affect this is still being understood.

### 1.7. Pancreatic Exocrine Insufficiency and the Intestinal Microbiome

PEI is thought to lead to changes in the composition of the gut microbiome, leading to a state of dysbiosis characterised by an imbalance in microbiome diversity [67,68,69,70]. Understanding how the microbiome and PEI are interconnected could aid the search for novel diagnostic and therapeutic targets and help improve outcomes in PDAC. Indeed, recent evidence indicates a role for the gut microbiome in the aetiology and progression of PDAC [19] and has also linked changes in the gut microbiome to outcome and treatment response [20,39,71,72,73]. The aim of this review is to summarise the current evidence linking PEI with microbiome dysbiosis, assess how microbiome composition may be influenced by PERT treatment, and look towards possible future diagnostic and therapeutic targets in this area.

## 2. The Interaction between Pancreatic Exocrine Function and Gut Microbiome

### 2.1. Effects of Pancreatic Exocrine Insufficiency on the Gut Microbiome

PEI is prevalent in patients with PC and there is emerging evidence that pancreatic exocrine function is closely linked to the composition of the gut microbiome [67,68,69]. Differences in the gut microbiome have been reported in patients with PEI, the main changes observed being a reduction in alpha-diversity, small intestine bacterial overgrowth and loss of ‘beneficial’ bacteria [67,68,69]. A large population-based study by Frost et al. gives significant insight into the importance of pancreatic exocrine function in regulating the gut microbiome [67]. This study compared the composition of the gut microbiome, via 16S rRNA gene sequencing of stool samples, to pancreatic function, using pancreatic elastase levels in over 1700 patients [67]. Their results showed that changes in the gut microbiome composition were much more closely linked to pancreatic exocrine function, rather than other commonly measured host factors such as age, dietary factors, diabetes or smoking [67]. This led the authors to conclude that pancreatic exocrine function appeared to be the most important host factor involved in shaping the human intestinal microbiome [67]. It is therefore likely that patients with pancreatic cancer, many of whom have PEI [42], will also have a deranged gut microbiome (Figure 1A) [19].

### 2.2. Pancreatic Regulation of Gut Microbiome

With increasing interest in the role that pancreatic exocrine function plays in the regulation of the gut microbiome, attention has turned towards the exact mechanisms by which pancreatic exocrine products influence gut bacteria. There are three main cell types in the adult pancreas: acinar cells that produce digestive enzymes; duct epithelial cells that produce a bicarbonate rich fluid which functions to carry digestive enzymes into the digestive tract and also to establish an optimal pH for enzyme function, and the cells of the Islets of Langerhans, which have an endocrine function, producing hormones such as glucagon and insulin [74,75]. In one large study, the role of the two main exocrine cell types (acinar and ductal) were investigated separately [67]. Acinar cell function was quantified with an immunochemical faecal elastase assay [76,77], and ductal cell function was assessed using secretin-stimulated magnetic resonance cholangiopancreatography (sMRCP) [67,78]. The results from this study demonstrated that acinar cell function was more closely related to gut microbiome composition than ductal cell function [67].

Acinar cells produce many different products, so determining which of these play a role in the composition of the gut microbiome has also been investigated. It has been shown that pancreatic secretions possess both bacteriostatic activity against common bacterial pathogens and fungistatic activity against Candida albicans [79]. However, this function of canine pancreatic secretions persisted when digestive enzymes were inhibited, which likely indicates that digestive enzymes are not the main factors that regulate the intestinal flora [67,79]. Apart from digestive enzymes, it has been shown that the pancreas produces antimicrobial peptides [67,80,81,82]. One such example of important antimicrobial peptides produced by the pancreas are cathelicidins, these are small proteolytically activated peptides shown to be active against bacteria, fungi and some viruses [83]. Murine studies have indicated that cathelicidin-related antimicrobial peptide (CRAMP) plays an important role in the regulation of the intestinal microbiome; when genetic modifications were made to reduce levels of CRAMP production in the murine pancreas, fatal bacterial overgrowth in the intestine occurred [81]. In human studies however, cathelicidin antimicrobial peptide (CAMP) levels in faeces have not been shown to be different in patients with PEI compared to healthy controls [67], this does not necessarily mean that this peptide is unimportant and may be due to sampling difficulties [67]. These products and their roles are summarised in Table 1.

Another mechanism by which it is thought PEI may lead to significant changes in the gut microbiome is by the oversupply of food products to the intestine, leading to the overgrowth of dominant bacteria [70]. In particular, it has been suggested that the increased availability of complex carbohydrates, seen in patients with PEI due to reduced digestive activity, may act as a substrate for certain communities within the gut microbiome [67]. Given that acinar cell activity has been closely linked to gut microbiome composition in humans [67], and the main function of these cells is to produce digestive enzymes, this seems like another plausible mechanism by which PEI may influence gut microbiome composition.

### 2.3. Gut Microbiome Changes from Pancreatic Exocrine Insufficiency in Different Disease States

Whilst there is credible evidence from both animal and human studies to suggest that PEI leads to significant changes in the gut microbiome, it is important to understand the impact that different disease processes leading to PEI in humans may have on the gut microbiome, such as chronic pancreatitis (CP) and pancreatic cancer (PC). For example, chronic pancreatitis (CP) is one of the main causes of PEI in humans and is known to be linked to changes in the gut microbiome [68,69,70,84], but it is often caused by excessive smoking or alcohol consumption, which in themselves may lead to changes in the gut microbiome. Indeed, a recent study in which the gut microbiome of chronic pancreatitis patients was investigated with 16S rRNA gene sequencing and compared to a matched population of healthy controls described some important findings [68]. The study by Frost et al. reported that not only did CP lead to significant changes in the gut microbiome, namely reduced microbial diversity and increased abundance of pathogenic bacteria, but more importantly these changes were independent of exocrine function (measured with stool elastase) [68]. This suggests that gut microbiome changes in pancreatic disease are not solely related to reduced pancreatic exocrine function.

At present, there is limited data on the changes in the gut microbiome related to PEI in PC patients. It is plausible that, as observed in CP patients [68], there are multiple factors influencing the gut microbiome composition in PC patients, beyond just PEI. For example, many PC patients experience cancer cachexia, characterised by systemic inflammation and nutritional depletion [53]. Links between the state of cancer cachexia and the gut microbiome have been drawn [85,86] and this means that changes in the gut microbiome in PC patients are likely to differ to those in CP patients, even though both experience PEI. As such, further study is required to understand the changes in the gut microbiome changes in different pancreatic disease processes such as CP and PC. Improved understanding of the changes to the gut microbiome in both CP and PC patients will inform future studies investigating therapeutics that aim to return the gut microbiome towards a healthier phenotype in these patients.

### 2.4. Effects of Pancreatic Enzyme Replacement on the Gut Microbiome

From the evidence outlined above, it appears that PEI is associated with gut microbiome dysbiosis, which in turn is thought to be associated with multiple adverse downstream effects. Therefore, it is desirable to identify strategies by which harmful effects of PEI on the gut microbiome may be mitigated and the microbiome returned towards a healthier state. One logical method in PC patients would be through pancreatic enzyme replacement therapy (PERT), which is frequently under prescribed, despite being recommended for all with pancreatic cancer by national practice guidelines [61,62]. In animal models, PERT has been shown to reverse changes to the microbiome caused by PEI [87,88]. Specifically, an increase in beneficial bacterial species, such as *Akkermansia muciniphila* and *Lactobacillus reuteri* [87,88], and a reduction in species associated with dysbiosis such as *Prevotella* and those of the phylum *Proteobacteria* [87]. This reestablishment of a healthy complex microbiome after dysbiosis has occurred (sometimes referred to as ‘rebiosis’ [69]) (Figure 1B), reduces intestinal inflammation, improves PEI-associated symptoms and mediates nutritional decline [69,88]. Therefore, it may be a further mechanism by which PERT acts to improve quality of life [57,58] and survival amongst pancreatic cancer patients [59,60] (Figure 2). That said, some animal evidence has suggested that the gut microbiome of animals treated with PERT still exhibits important differences to healthy controls [89]. As a result, further study is required to evaluate to what degree PERT can return the gut microbiome of PC patients with PEI towards a healthier composition and whether this translates into improved outcomes.

## 3. Effects of the Intestinal Microbiome in Pancreatic Cancer Patients

### 3.1. Intestinal Microbiome and Clinical Outcomes in Pancreatic Cancer

Associations between microbiome composition and clinical outcome for patients with PDAC are now being observed. Differences in intestinal microbiome composition between patients with short-term survival (STS) and long-term survival (LTS) have been demonstrated with 16S rRNA gene sequencing [21,39]. An increasing abundance of *Acinetobacter* was identified in the highly aggressive ‘basal-like’ subtype of pancreatic ductal adenocarcinoma (PDAC) using metagenomic sequencing of resected tumours [90]. However, in the absence of mechanistic insights, these findings remain associative rather than causative.

### 3.2. Extraintestinal Effects of the Gut Microbiome in Pancreatic Disease

The effects of intestinal microbiome dysbiosis have been shown to extend beyond the intestinal tract in patients with pancreatic disease. In acute and chronic pancreatitis, a frequent clinical issue is the colonisation of pancreatic necrosis and fluid collections with pathogenic bacteria. Evidence suggests that numbers of beneficial bacterial species such as *Faecalibacterium* [91,92] and *Fusicatenibacter* [93] are reduced in patients with CP [68]. These bacterial species are thought to play an anti-inflammatory role in the colon and preserve the intestinal barrier through the production of short-chain fatty acids and lactate [91,92], reducing the chance of extraintestinal translocation of pathogenic bacteria [93,94]. As a result, gut microbiome changes can be directly related to the severity of the disease [68,95].

Importantly, for pancreatic cancer patients, it is now appreciated that pancreatic tumours themselves harbour a population of bacteria, referred to as the tumour microbiome [21,73,96,97], which has been shown to influence the prognosis of PDAC [21,39,98]. Relevant to this article, it has been shown that the gut microbiome influences the composition of this tumour microbiome in pancreatic cancer [39]. The exact mechanism for this remains debated, but possible routes include retrograde translocation from the gut, via the duodenum and bile/pancreatic ducts, spread from the portal circulation or mesenteric lymph nodes (Figure 2) [20,21,22]. The importance of the tumour microbiome is attracting increasing interest [96,99] and has been shown to differentiate between different phenotypes of pancreatic cancer, as well as influencing clinical outcome [39,90]. For example, some human pancreatic tumours are colonised by *Gammaproteobacteria*, a subset of which possess an isoform of cytidine deaminase, which is capable of metabolising gemcitabine into an inactive form [21] and, in so doing, increases the tumour’s chemoresistance and therefore reduces treatment efficacy [100,101]. Evidence of clinical benefit in modulating the tumour microbiome has also been demonstrated in a recent large cohort study which showed that peri-chemotherapy antibiotics in metastatic PDAC patients improved cancer-specific survival [102]. Interestingly, a survival benefit was seen only in patients receiving gemcitabine, not fluorouracil, adding to the growing body of evidence that bacteria-mediated gemcitabine resistance is prevalent in PDAC patients [20] and could represent a target for future therapies.

### 3.3. Tumour Microbiome and Immune Infiltration

A further route by which it is thought that the tumour microbiome influences the prognosis of PDAC includes local immunosuppression and subsequent accelerated oncogenesis [103,104,105,106]. A relationship between the tumour microbiome and tumour immune infiltration has been demonstrated [103] with an increase in infiltration of CD8+ T cells reported following ablation of the tumour microbiome with antibiotics [103]. Furthermore, transfer of selected bacteria from PDAC tissue accelerated tumorigenesis [103]. This highlights a possible mechanism by which a more favourable gut microbiome, which is observed through appropriate treatment of PEI with PERT, may influence cancer-related outcomes in PDAC patients. Future studies should therefore seek to compare the gut and tumour microbiome composition to immune infiltration of tumours, for example via immunohistochemistry, to further investigate this relationship.

## 4. Diagnostic and Therapeutic Applications for Greater Understanding of the Gut Microbiome in Pancreatic Cancer Patients

### 4.1. Use of Intestinal Microbiome Testing for Diagnosis of Pancreatic Exocrine Insufficiency

If changes in the microbiome are observed in PEI, then it may also be possible to diagnose this using intestinal microbiome sequencing from stool samples. The incidence of PEI pre-operatively is challenging to assess, and no ideal diagnostic test currently exists. The secretin test and coefficient of fat absorption (CFA) are accurate but expensive and unpleasant, whilst faecal elastase (FE-1), although suitable for routine use, has poor sensitivity in mild PEI [107]. Traditionally, a combination of patient reported symptoms, such as steatorrhoea, and faecal elastase (FE-1) have been used to diagnose PEI. However, the diagnostic accuracy of FE-1 has been questioned, especially amongst those who have undergone surgical resection where the increased ratio of faecal fat to FE-1 reduces the sensitivity of the test [107,108]. The ^13^C mixed triglyceride breath test is emerging as a promising diagnostic tool [109], but expense and time limit its use to specialist or research centres only [109,110]. Therefore, if changes in the microbiome in patients with PEI are further understood, these could be measured and quantified to be used as a new diagnostic test.

### 4.2. Titration of Pancreatic Enzyme Replacement Therapy

Given the difficulties in diagnosing PEI in routine clinical practice, monitoring the response to PERT is arguably even more challenging. Whilst UK NICE recommend consideration of PERT treatment for all patients following pancreatic resection [61,62] without the need for a positive diagnostic test, it can be challenging to ensure that each patient is taking the correct dose. If changes in the intestinal microbiome with PERT treatment in PEI patients are quantifiable, then it may be possible to monitor long-term trends with PERT treatment and make dose changes accordingly using stool sample sequencing.

### 4.3. Positive Effects of Pancreatic Enzyme Replacement Therapies on the Intestinal Microbiome

As described above, early evidence suggests that by modulating the gut microbiome it may be possible to alter the composition of the tumour microbiome. This has been demonstrated in animal studies using faecal microbiota transplantation (FMT) [39,73]. Using the knowledge that differences in intestinal microbiome composition between short-term survival (STS) and long-term survival (LTS) patients have been demonstrated with 16S rRNA gene sequencing [21,39]; researchers sought to modulate the intestinal microbiome with FMT. Interestingly, it has been shown that via FMT from STS and LTS patients into mice, it is possible to modulate the tumour microbiome and thus affect tumour growth [39]. This could also link to some possible benefits of PERT as, if treatment is able to return the intestinal microbiome of pancreatic cancer patients towards a healthy composition, this may positively influence the tumour microbiome. Given that *Acinetobacter* abundance has been associated with aggressive ‘basal-like’ tumours [90], it is noteworthy that this can be reduced in the gut microbiome of a porcine PEI model reduced after administration of PERT [87]. The potential ability of PERT to modulate the intestinal microbiome, and consequently the tumour microbiome, may be an important determinant of clinical outcome and is worthy of more thorough investigation.

## 5. Conclusions and Future Directions

It is likely that the gut microbiome of pancreatic cancer patients with PEI exhibits profound dysbiosis and that this may potentially be reversible with PERT. However, further research is needed to conclude if this is indeed the case. Future research should seek to elucidate the specific changes that occur in PC patients with PEI as these are likely to be different to gut microbiome changes in patients with other, benign causes of PEI such as cystic fibrosis or CP. It would also be important to investigate whether the gut microbiome of PC patients who have PEI treated with PERT experience a return of gut microbiome towards that of healthy controls. It is also important to acknowledge that supplements, such as probiotics as well as faecal microbiota transplantation, can influence the composition of the gut microbiome and therefore may also be beneficial in PC [111].

In addition, the downstream effects of gut microbiome dysbiosis in PC patients with PEI need to be further explored and understood. For example, future studies should seek to compare the gut microbiome with the tumour microbiome in PC patients with and without PEI. Similarly, the interaction between the gut microbiome and the immune system in this cohort should be further investigated. This could be achieved by measuring the peripheral immune state in PC patients with and without PEI, or before and after PERT treatment. Equally, the immune infiltration of resected tumours could be compared to the gut microbiome of PC patients and their degree of exocrine function and/or PERT treatment.

It is also important to investigate the extent to which the gut microbiome influences chemosensitivity in PC patients, and whether this is influenced by the degree of PEI or PERT. This may be achieved by monitoring treatment response to neoadjuvant chemotherapy via Ca19-9 levels, radiological response or tumour regression grading of resected specimens, and correlating this to exocrine function and PERT treatment.

In summary, it has been recognised that untreated PEI is harmful in PC patients and that PERT treatment leads to improved outcomes. It has been proposed that this is likely due to improved nutritional status, thus allowing more patients to undergo physiologically demanding treatments, but it is possible that treatment of PEI with PERT in PC patients leads to a more favourable gut microbiome. This in turn may modulate the tumour microbiome, thereby directly influencing tumour growth and the quality of the tumour-specific immune response. The effect of PEI and PERT on the gut microbiome of pancreatic cancer patients is a vital area for future study and may open the door to much needed new diagnostic and therapeutic options for this challenging condition.

## Figures and Tables

**Figure 1 cancers-15-05140-f001:**
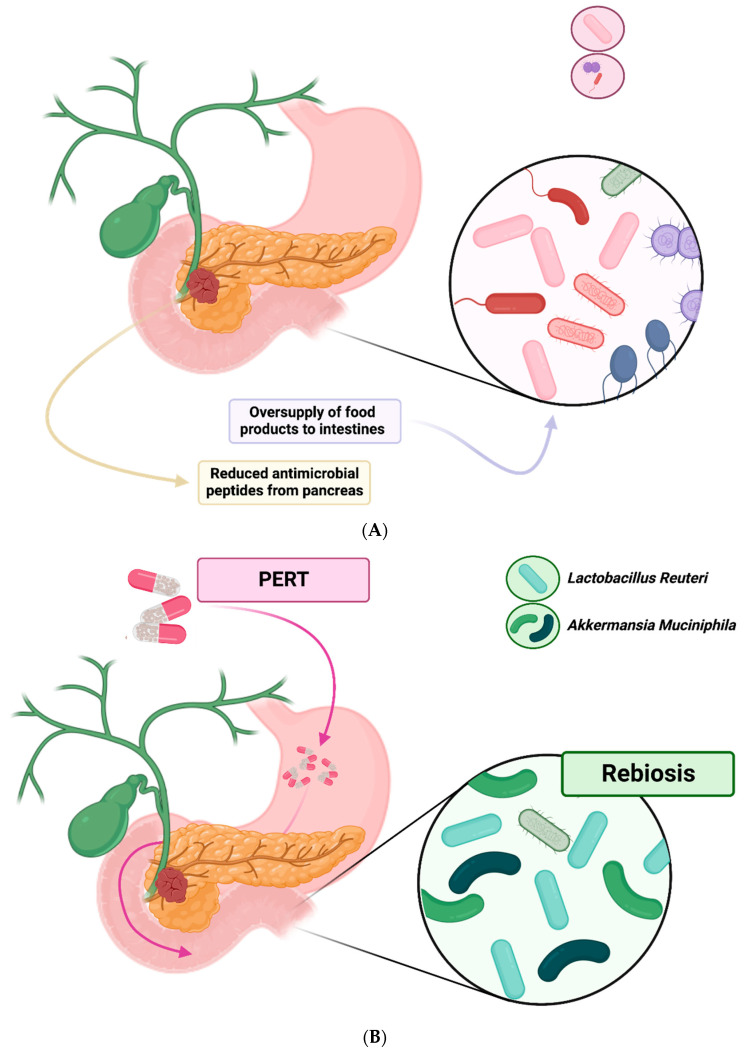
(**A**) mechanisms of dysbiosis in PEI; (**B**) mechanisms of rebiosis in PEI.

**Figure 2 cancers-15-05140-f002:**
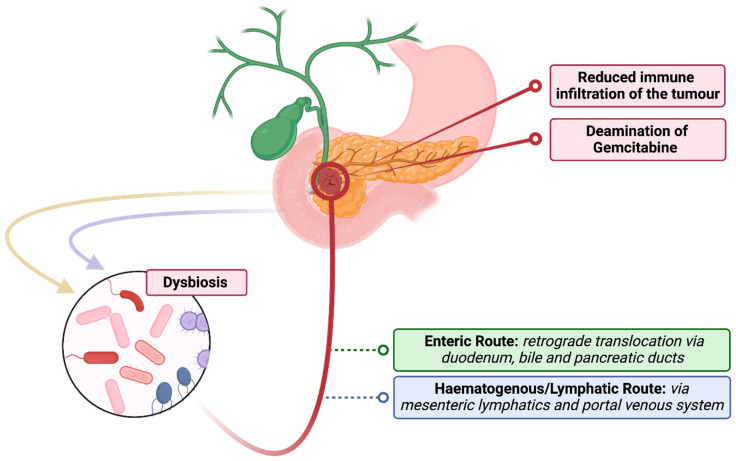
Links between gut microbiome and pancreatic tumour microbiome.

**Table 1 cancers-15-05140-t001:** Summary of exocrine pancreatic products and their effect on the gut microbiome.

Exocrine Pancreatic Product	Example	Source Pancreatic Cell Type	Function	Effect on Gut Microbiome
Bicarbonate rich fluid	-	Duct epithelial cell	Carry pancreatic enzymes into small intestine. Optimise pH for enzyme action.	sMRCP indicates negligible role
Digestive enzymes	Lipase, protease, amylase	Acinar cell	Digest food products	Oversupply of food products leads to significant changes especially complex carbohydrates. Inhibition in canine models did not affect composition of microbiome.
Antimicrobial peptides	CAMP and CRAMP	Acinar cell	Bacteriostatic properties	Murine studies indicate a regulatory role for the gut microbiome. Human studies remain inconclusive due to sampling issues

sMRCP = secretin-stimulated magnetic resonance cholangiopancreatography CAMP = cathelicidin antimicrobial peptide; CRAMP = cathelicidin-related antimicrobial peptide.

## Data Availability

Not applicable.

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
