# Peer review of "Pancreatic Exocrine Insufficiency and the Gut Microbiome in Pancreatic Cancer: A Target for Future Diagnostic Tests and Therapies?"

_cancers, 2023, doi:10.3390/cancers15215140_

Round 1
Reviewer 1 Report
The article is well written and addresses a topical issue of broad interest. As is known, numerous scientific studies assign an importance to the intestinal microbiome in reducing infections after colorectal surgery, in regulating the progression of inflammatory bowel diseases and in the evolution of chronic liver disease. The correlation with the exocrine function of the pancreas represents a new interaction that deserves careful interest. Regarding the article, I would like to point out that some epidemiological data reported are inadequate, in particular the reference to the prognosis and five-year survival of surgical treatment of pancreatic cancer which is reported to be less than 5%. Recent articles confirm that surgical resection of pancreatic ductal carcinoma, in the early stage, represents the only treatment option with five-year survival in 30% of cases. The further addition of adjuvant chemotherapy can improve this outcome (1). Another aspect that should be considered after duodenum pancreatectomy is the method of reconstruction of intestinal transit which, in our opinion, represents an important data for the evaluation of the exocrine function of the pancreas. As demonstrated by some of our articles, the preservation of the first jejunal loop anastomized to the stomach, allows us to maintain the receptors of the neuroendocrine cells present on the first jejunal loop capable of stimulating pancreatic exocrine secretion and therefore allowing a good balance on the absorption of lipids and proteins.(2, 3,4) This data, if taken into consideration, could constitute a further element of evidence for evaluating the role of the intestinal microbiome. In conclusion I believe that the article, with these minor revisions, could be published. The cited articles are attached:
(1) Predective biomarkers for personalized approach in resectable pancreatic cancer. Frontiers in Surgery 2022. (2) Monitoring fibrosis of the pancreatic remnant after a pancreaticoduodenectomy with dynamic MRI: are the results independent of the adopted reconstructive technique? J. Surg. 2010 Res. 164: e49-e52.) (3) Endocrine cells distribution in human proximal small intestine: an immunohistochemical and morphometrical study - Itlian Journal of Anatomy and Embriology (IJAE) Vol . 121, n . 1: 112-121, 2016 (4) Occurrence of pathological abdominal fat distribution after pancreaticoduodenectomy at long-term follow-up: a single-center experience PANCREAS 2021
Author Response
The article is well written and addresses a topical issue of broad interest. As is known, numerous scientific studies assign an importance to the intestinal microbiome in reducing infections after colorectal surgery, in regulating the progression of inflammatory bowel diseases and in the evolution of chronic liver disease. The correlation with the exocrine function of the pancreas represents a new interaction that deserves careful interest.
Thank you for your kind comments.
Regarding the article, I would like to point out that some epidemiological data reported are inadequate, in particular the reference to the prognosis and five-year survival of surgical treatment of pancreatic cancer which is reported to be less than 5%. Recent articles confirm that surgical resection of pancreatic ductal carcinoma, in the early stage, represents the only treatment option with five-year survival in 30% of cases. The further addition of adjuvant chemotherapy can improve this outcome (1).
Thank you for your comment. In section 1.3 we refer to the overall survival for all patients presenting with PDAC which, giving that around 80% are unresectable, is around 5%. However we acknowledge that this was not clear so we have amended the sentence and have added the reference you have suggested.
Another aspect that should be considered after duodenum pancreatectomy is the method of reconstruction of intestinal transit which, in our opinion, represents an important data for the evaluation of the exocrine function of the pancreas. As demonstrated by some of our articles, the preservation of the first jejunal loop anastomized to the stomach, allows us to maintain the receptors of the neuroendocrine cells present on the first jejunal loop capable of stimulating pancreatic exocrine secretion and therefore allowing a good balance on the absorption of lipids and proteins.(2, 3,4) This data, if taken into consideration, could constitute a further element of evidence for evaluating the role of the intestinal microbiome.
Thank you for your important points, we have added these into the introduction with the accompanying references,
In conclusion I believe that the article, with these minor revisions, could be published.
Thank you for your kind comments and feedback which has enhanced the article.
Reviewer 2 Report
The authors prepared a well-structured review that needs a revision to include information about pancreatic enzymes, their function and what effect each enzyme (or non-enzymatic products) have on dysbiosis. Please make a table of all enzymes, products, how it effects or might affect bacterial composition. These tables will allow readers to save time and enhance manuscript experience by rapid data presentation.
Please characterize "1.7 Pancreatic Exocrine Insufficiency". The term PEI is often used without explanation of enzymes produced by pancreas. List all the enzymes, their function and what products are in excess in intestine/malabsorbed deficiencies with exocrine insufficiency. The readers will appreciate a complete set of information. I recommend to make a table with a list of enzymes and what each enzyme deficiency leads to, how it affects each bacteria population
Figure 1. A – mechanisms of dysbiosis in PEI. - diagram lists "oversupplly of food products and reduced antimicrobial peptides. Please list all food products and all antimicrobial peptides. This is a wasted opportunity to introduce all factors in an easy to see graphical format that are later dispersed throughout the manuscript, eg, cathelicidin, CRAMP and etc
Please characterize "1.6 Pancreatic Enzyme Replacement Therapy" What enzymes are replaced and at what schedule. Pancreas produces a diversity of enzymes and its not enough to blanket term them as PEI without explaining each enzyme function and its effect on bacteria either directly, or through a malabsorbed metabolite.
Author Response
The authors prepared a well-structured review that needs a revision to include information about pancreatic enzymes, their function and what effect each enzyme (or non-enzymatic products) have on dysbiosis. Please make a table of all enzymes, products, how it effects or might affect bacterial composition. These tables will allow readers to save time and enhance manuscript experience by rapid data presentation.
Thank you we have added a table of the main products produced by the pancreas and their roles to summarise this as requested.
Please characterize "1.7 Pancreatic Exocrine Insufficiency". The term PEI is often used without explanation of enzymes produced by pancreas. List all the enzymes, their function and what products are in excess in intestine/malabsorbed deficiencies with exocrine insufficiency. The readers will appreciate a complete set of information. I recommend to make a table with a list of enzymes and what each enzyme deficiency leads to, how it affects each bacteria population
Thank you we have added a definition of PEI and a landmark reference at the start of section 1.5 and hope that is satisfactory. Exact details of how each product affects the gut microbiome composition is still debated but we have included the most relevant information in the new table and in the text. We hope that this is satisfactory.
Figure 1. A – mechanisms of dysbiosis in PEI. - diagram lists "oversupplly of food products and reduced antimicrobial peptides. Please list all food products and all antimicrobial peptides. This is a wasted opportunity to introduce all factors in an easy to see graphical format that are later dispersed throughout the manuscript, eg, cathelicidin, CRAMP and etc
Thank you for your comment, In section 2.2 we reference studies which indicate that oversupply of complex carbohydrates in particular are thought to be the main contributing factor to gut microbiome dysbiosis in PEI. Beyond this there still remains debate about the exact effect that different dietary components have on the gut microbiome. We hope that is satisfactory. Similarly, there are many catheledicin-related antimicrobial peptides but the exact role of each of these in dysbiosis of the gut microbiome in PEI remains debated and in need of further research. We hope that is satisfactory.
Please characterize "1.6 Pancreatic Enzyme Replacement Therapy" What enzymes are replaced and at what schedule. Pancreas produces a diversity of enzymes and its not enough to blanket term them as PEI without explaining each enzyme function and its effect on bacteria either directly, or through a malabsorbed metabolite.
Thank you, PERT is orally administered capsules containing manufactured enzymes (lipase, protease and amylase). Dosing independent on patient symptoms and consition and is outlined in the specialist guidance referenced in section 1.6. We have defined PERT in the opening sentence of this paragraph and hope that is satisfactory
Reviewer 3 Report
his review focus on the current evidence linking pancreatic exocrine insufficiency (PEI) with microbiome dysbiosis, assess how microbiome composition may be impacted by pancreatic enzyme replacement therapy (PERT) treatment, and look towards possible future diagnostic and therapeutic targets in this area. This topic is interesting. This paper was well organized but some minor mistakes need to be corrected before acceptance.
1. It is suggested to add a comma after [4-7] on line 59.
2. It is suggested to revise the sentence on line 76.
3. It is suggested to revise the number format of references on line 92 and line 175.
4. It is suggested to revise “mechanisms” as “Mechanisms” on line 185 and line 280 like line 282.
his review focus on the current evidence linking pancreatic exocrine insufficiency (PEI) with microbiome dysbiosis, assess how microbiome composition may be impacted by pancreatic enzyme replacement therapy (PERT) treatment, and look towards possible future diagnostic and therapeutic targets in this area. This topic is interesting. This paper was well organized but some minor mistakes need to be corrected before acceptance.
1. It is suggested to add a comma after [4-7] on line 59.
2. It is suggested to revise the sentence on line 76.
3. It is suggested to revise the number format of references on line 92 and line 175.
4. It is suggested to revise “mechanisms” as “Mechanisms” on line 185 and line 280 like line 282.
Author Response
his review focus on the current evidence linking pancreatic exocrine insufficiency (PEI) with microbiome dysbiosis, assess how microbiome composition may be impacted by pancreatic enzyme replacement therapy (PERT) treatment, and look towards possible future diagnostic and therapeutic targets in this area. This topic is interesting. This paper was well organized but some minor mistakes need to be corrected before acceptance.
Thank you for your kind comment
- It is suggested to add a comma after [4-7] on line 59.
Added, thank you
- It is suggested to revise the sentence on line 76.
Updated, thank you
- It is suggested to revise the number format of references on line 92 and line 175.
Updated, thank you
- It is suggested to revise “mechanisms” as “Mechanisms” on line 185 and line 280 like line 282.
Updated, thank you
Reviewer 4 Report
This is a review article on PEI and gut microbiota in relation to pancreatic cancer. The paper is comprehensive.
1. One of the characteristics of pancreatic cancer is its association with obesity and diabetes. Please discuss sarcopenic obesity and gut microbiota with pancreatic cancer.
2. There are some clinical trials of PERT in cases with pancreatic cancer.
3. Please also add the role of probiotics, prebiotics etc. in this setting.
4. The title can be misleading that test for PEI can diagnose pancreatic cancer.
Author Response
This is a review article on PEI and gut microbiota in relation to pancreatic cancer. The paper is comprehensive.
- One of the characteristics of pancreatic cancer is its association with obesity and diabetes. Please discuss sarcopenic obesity and gut microbiota with pancreatic cancer.
Thank you for your comment. We have mentioned in section 1.5 that PEI is associated with sarcopenia and added some important references here to further guide readers. We have also added an extra sentence to clarify this further and hope that is satisfactory.
- There are some clinical trials of PERT in cases with pancreatic cancer.
Thank you, we have referenced some important studies in section 1.6 and hope that is satisfactory
- Please also add the role of probiotics, prebiotics etc. in this setting.
Thank you for your comment. There is certainly a role for supplements such as probiotics in this setting and we have detailed these in a separate review. We have added this to the future direction and conclusion section given that it is an important subject. However, we feel that a detailed discussion of these supplements is beyond the scope of the current review. We hope that is satisfactory. We have added a reference to further guide readers.
- The title can be misleading that test for PEI can diagnose pancreatic cancer.
Thank you for your comment. Whilst in the very early stages, as we discuss in section 1.4, some studies have identified signatures for PDAC in the gut microbiome early in the disease process. With further investigation, this could result in identification of an early diagnostic biomarker. We hope that is satisfactory.